# Bioelectrical Impedance Versus Biochemical Analysis of Hydration Status: Predictive Value for Prolonged Hospitalisation and Poor Discharge Destination for Older Patients

**DOI:** 10.3390/healthcare9020154

**Published:** 2021-02-03

**Authors:** Adrian D. Wood, Gillian D. Edward, Kirsten Cumming, Mohannad W. Kafri, Roy L. Soiza, Lee Hooper, John F. Potter, Phyo K. Myint

**Affiliations:** 1Ageing Clinical & Experimental Research Team, Institute of Applied Health Sciences, University of Aberdeen, Aberdeen AB24 3FX, UK; a.d.wood@abdn.ac.uk (A.D.W.); g.d.edward.12@aberdeen.ac.uk (G.D.E.); kirsten.cumming@nhs.scot (K.C.); mwkafri@gmail.com (M.W.K.); roy.soiza@nhs.scot (R.L.S.); 2Department of Nutrition & Dietetics, Birzeit University, Birzeit P.O. Box 14, Palestine; 3Department of Medicine for the Elderly, Aberdeen Royal Infirmary, NHS Grampian, Aberdeen AB25 2ZN, UK; 4Norwich Medical School, University of East Anglia, Norwich NR4 7TJ, UK; l.hooper@uea.ac.uk (L.H.); john.potter@uea.ac.uk (J.F.P.)

**Keywords:** dehydration, bioelectrical impedance, intracellular, extracellular, blood urea nitrogen, creatinine

## Abstract

Dehydration is prevalent in hospitalised patients and is associated with increased morbidity and mortality, particularly among the elderly (≥65 years). We aimed at comparing the performance of intracellular water to extracellular water ratio (ICW/ECW), calculated through a bioelectrical impedance analysis (BIA) of blood urea nitrogen, with the creatinine ratio (BUN/Cr) to predict poor outcomes in a cohort of prospectively identified patients. Data were combined from a cohort of elderly patients (≥65 years) admitted to hospital with fragility fracture (*n* = 125) and older adults aged ≥50 years admitted to hospital with stroke (*n* = 40). The association between hydration status and study outcomes (unfavourable discharge destination (rehabilitation, another ward, or death) and prolonged hospitalisation (>10 days)) was examined using logistic regression. The overall diagnostic accuracy of each hydration status measurement was assessed using the area under the receiver operating characteristic (ROC) curve. In 165 participants (mean age (SD) of 76.7 (9.2) years), an ICW/ECW ratio below the 25th percentile was associated with increased odds of poor discharge destination (OR (95% CI) = 4.25 (1.59–11.34)). Neither the relationship between the BUN/Cr ratio and prolonged stay nor discharge destination was significant. A BIA could be used utilised in conjunction with biochemical measurements to inform patient prognosis.

## 1. Introduction

Disturbances of fluid and electrolyte balance are common in clinical practice. Age-related pathophysiological changes in the handling of fluid and electrolytes in older persons (>65 years) predisposes this demographic to an increased risk of morbidity and mortality from dehydration [1]. Ageing-associated loss of body water due to reduction in skeletal muscle mass, increased adiposity, and declining renal function, and thirst perception contributes to disruption of normal fluid homeostasis [2,3]. It has been reported that dehydration is present in up to 40% of older patients, aged ≥65 years admitted to hospital in the UK [4]. This could be an underestimation due to the difficulty of assessing dehydration [5,6].

Dehydration can be assessed clinically. Tongue dryness has previously been shown to be more strongly associated with poor hydration status (demonstrating 64% sensitivity and 62% specificity) than other clinical signs and symptoms such as thirst, orthostatic systolic blood pressure drop, and sternal skin turgor [7]. However, the clinical diagnosis may be subjective, and therefore could be prone to error [8]. A more objective measurement of dehydration can be achieved by employing laboratory tests such as blood urea nitrogen (BUN) and creatinine (Cr), both of which are also markers of renal function.

Elevated BUN to Cr (BUN/Cr) ratio (≥15) has been shown to be associated with a poor clinical outcome (death, placement in hospice or nursing home) in a cohort of acute ischaemic stroke patients (OR (95% CI) = 2.2 (1.2–4.0)) [9]. Dehydration may also be determined from the intracellular water (ICW) to extracellular water (ECW) ratio (ICW/ECW). In healthy persons, bodily fluids are maintained in an ICW/ECW ratio of 2:1. A 1% rise in plasma osmolality (depleted ICW) triggers the thirst and the release of anti-diuretic hormone which acts to retain body water [10].

Since severe water loss dehydration requires immediate medical intervention due to the significant health risks posed by hypovolaemia and associated electrolyte disturbances, a fast and reliable method of detecting dehydration could prove to be incredibly valuable. A bioelectrical impedance analysis (BIA) is a quick, relatively inexpensive, and non-invasive method that estimates both total body water (TBW) and ECW, used for calculation of the ICW, and hence body composition and volume, measured by passing an electric current through the body [11].

Against this background, we aimed at comparing the ICW/ECW ratio (estimated from bioelectrical impedance analysis) and the BUN/Cr ratio in predicting the likelihood of prolonged hospitalisation and discharge to an unfavourable destination (rehabilitation, transfer to another ward or hospital, or in-patient death) in a cohort of prospectively identified patients with fragility fracture or stroke.

## 2. Materials and Methods

### 2.1. Study Design and Participants

Data were drawn from two patient cohorts. The first recruited cohort was elderly patients (aged >65 years) who were consecutively admitted to the Aberdeen Royal Infirmary, Scotland, UK. A further 40 prospectively identified stroke patients aged >50 years were recruited from the Norfolk and Norwich University Hospital, UK, for whom BIA data were also available.

Inclusion criteria for both cohorts comprised patients with the capacity to consent. Consecutively admitted fragility fracture patients were included if they were over 65 years of age. Consecutively admitted stroke patients were included if they were over 50 years of age. Participants were excluded if they were unable to give informed consent. Additional exclusion criteria applied to stroke patients were severe stroke (National Institute of Health Stroke Scale (NIHSS) score >30) [12], co-existing terminal illness, or expected survival of less than 48 h.

Written informed consent was obtained from all participants. Ethical approvals for fracture and stroke participants were, respectively, obtained from the North of Scotland (ref 12/SS/0209) and Cambridgeshire Research Ethics Committees (10/H0304/18). Our study protocol conformed to the ethical guidelines of the 1975 Declaration of Helsinki.

### 2.2. Outcome Measurements and Covariates

The BIA was performed according to the availability of measuring equipment. In fracture patients, a Quantum/S analyser was utilized with impedance measured at 50 kHz. In the stroke patients, a multi-frequency BIA equipment (Maltron BioScan, 920-2, Maltron International Co. Essex, UK) was utilised. In all participants, electrodes were attached to the upper and lower limbs (one electrode on each). The BIA measurements were taken on admission to determine TBW, ECW, and ICW values. The ICW/ECW ratio was calculated from these values. The detailed methods of the measurements and timing of the measurements have been previously described [13,14].

Routine blood specimens were collected within 48 h of hospital admission to determine the BUN/Cr ratio. The length of hospitalisation was recorded for each participant. Prolonged hospitalisation was defined as greater than 10 days (median length of stay). The destination at hospital discharge was recorded as follows: home, rehabilitation, transfer to another ward or hospital, and death. For this study, poor discharge destination was defined as discharge to a location other than home. Data were also collected on age, sex, weight, fracture type, and stroke classification (total anterior circulation stroke, partial anterior circulation stroke, posterior circulation stroke, and lacunar stroke).

### 2.3. Statistical Analysis

Data were analysed using the Statistical Package for Social Sciences (SPSS) 25.0 (SPSS, Chicago, IL, USA). Descriptive statistics were generated for all participants. Then, the relationship between hydration status and study outcomes was examined. Multivariable logistic regression models were employed to determine odds ratios for prolonged hospitalisation and poor discharge destination in patients with an ICW/ECW ratio below the 25th percentile and in patients with a BUN/Cr ratio of >15 (adjusted for age and sex). Factors for inclusion into our statistical models were agreed a priori. The diagnostic parameters, sensitivity, specificity, positive predictive value, and negative predictive value for study outcomes were estimated for each method of assessing hydration status as compared with the reference. The overall diagnostic accuracy was measured using the area under the receiver operating characteristic (ROC) curve.

## 3. Results

### 3.1. Participant Characteristics

From a total of 214 fracture patients who were approached initially, 66 patients were unable to give informed consent and 19 patients declined to participate. No fracture was found in two patients and a further two patients withdrew due to post-operative pain. Of the 40 stroke patients deemed suitable to participate in our study, all consented to take part. The present study, therefore, included 165 participants (mean age (SD) = 76.7 ± 9.2 years). The majority of participants were female (69.7%). Participant characteristics are summarised in Table 1.

### 3.2. Study Outcomes

The median length of hospital stay was six days. Of the 165 participants, 67 patients were discharged to home, 63 patients were discharged to rehabilitation, 26 patients were transferred to another ward or hospital (at their usual town of residence), and four patients died during their hospital stay (Table 2).

Table 3 shows the results from the multivariable logistic regression analyses examining the relationship between measurements of hydration and study outcomes. An ICW/ECW ratio below the 25th percentile (<0.89) was independently associated with increased odds for poor discharge destination (OR (95% CI) = 4.25 (1.59–11.34)). A BUN/Cr ratio of >15 (*n* = 19) was not associated with prolonged hospital stay, nor poor discharge destination. An ICW/ECW ratio below the 25th percentile showed no association with prolonged hospitalisation. The area under the ROC curve (AUC) for the logistic regression model based on the ICW/ECW ratio was AUC 0.74 (CI 0.67–0.82, *p* < 0.001) as compared with AUC 0.61 (CI 0.52–0.70, *p* = 0.02) for the BUN/Cr ratio. Thus, the discriminatory ability of the latter measurement was lower.

## 4. Discussion

In this prospectively identified cohort of patients admitted to hospital with fragility fracture or stroke, our data show that poor hydration status assessed using the ICW/ECW ratio is associated with unfavourable discharge destination (to a location other than home). The ICW/ECW ratio measured using a BIA is fast, non-invasive, and could be utilised in clinical settings to predict outcome for older patients and appears to be a better prognostic indicator than the BUN/Cr ratio. Implementation of this simple test at the bedside may have the potential to detect impending dehydration in older persons, and therefore clinicians could intervene in a timely manner to prevent the adverse health impacts of mild water loss or to reduce the incidence of more severe dehydration which can have more devastating health consequences.

A BIA has previously been employed as a tool to assess patient prognosis in a variety of cohorts. For example, elevated extracellular water to total body water ratio is associated with increased odds of hospital readmission in heart failure patients [15]. Elevated intracellular water is an independent predictor of poorer survival in cancer patients [16].

The extracellular water to total body water ratio can vary according to hepatitis virus type and may be useful for predicting liver fibrosis [17].

We propose that a BIA could be used to guide patient management rather than as a gold standard measurement of dehydration. Although it has previously been shown to be more accurate at determining hydration status than a clinical examination [14], we would caution its use as a stand-alone test to assess water-loss dehydration. Indeed, it has been suggested that a variety of methods for assessing dehydration in hospitalised persons (such as fluid intake, urine volume, colour, or specific gravity, heart rate, dry mouth, and feeling thirsty) are not useful in isolation, particularly in elderly persons [18]. Although hydration status assessed by a BIA appears to be better associated with poor patient outcome, this may simply be attributable to the fact that a BIA and biochemical analyses are different modalities which measure different constructs. It is possible that a BIA may reflect chronicity of poor hydration status (not easily reversible during admission), whereas biochemical analyses may reflect a more acute picture and hence be influenced by treatments received. It may be useful to combine methods of assessing dehydration to improve the diagnostic accuracy, and therefore utilization of a BIA in the hospital setting may be a useful adjunct to traditional biochemical analyses. Purchase cost ranges from 3000 to 5000 pounds, and training on how to use the machine and upload data could be undertaken within a relatively short time frame (approximately 2 h for a group training session).

Although our study was undertaken in an acute hospital setting, it is possible that BIA could be a useful tool to inform prognosis in other clinical settings where routine laboratory measurements are less accessible. Standard blood tests on hospital admission commonly include urea and creatinine measurements aMnd these values are regularly available to hospital clinicians. It is possible that the use of BIA may be particularly informative within rural settings, where access to clinical resources is limited. This would require evaluation in future research.

We acknowledge certain study limitations. The ICW/ECW ratio was measured using BIA only once. Previous work has shown BIA to be a useful tool for assessing hydration when taking serial measurements in the same person, thereby reducing intra-individual variability [19], whilst assessment in multiple persons is likely less accurate. Outcomes were determined in two different patient cohorts potentially biasing our results, although statistical models were adjusted for age and sex. However, there are a number of other potentially confounding covariates such as comorbidity status, medication use, and lifestyle factors which we were unable to adjust for in our analysis as we did not collect such information as part of the study. BIA was measured in fracture patients and stroke patients using different analytical equipment. Thus, it could be argued that our findings may be limited by differences in measurement sensitivity across different instrumentation (although supplemental analysis showed comparable AUC for different methodologies). Whilst our data show that in the population group included in this study, the BUN/Cr ratio cannot be reliably employed to predict patient outcomes, this does not rule out its value in different demographic groups, such as younger patients or those with chronic disease. Indeed, our study cohort was made up of acutely unwell patients, and thus our results may not be generalisable to other patient groups. It is important to note that formal power calculations for this study were not performed, thus, our study may be limited in terms of power.

## 5. Conclusions

Implementation of the ICW/ECW ratio, an objective dehydration marker, on admission in hospitals alongside subjective clinical measurements or other calculated values could help with early prognostication to identify older patients (over 50 years of age) who are at risk of adverse outcomes by detecting dehydration as early as possible. Further research is required to establish whether this ratio can be used in other age and patient groups. Successfully managing dehydration has the potential to reduce prolonged hospitalization and improve patient outcomes by preventing adverse effects associated with severe dehydration.

## Figures and Tables

**Table 1 healthcare-09-00154-t001:** Participant characteristics.

Continuous Data	All	Fracture	Stroke
Age, years	76.7 (9.2)	78.7 (8.0)	70.3 (9.9)
Urea mmol/L	7.9 (5.3)	8.3 (5.6)	6.7 (4.3)
Creatinine µmol/L	79.5 (30.1)	77.9 (30.4)	84.9 (31.2)
BUN/Cr ratio	11 (4.5)	11.6 (4.3)	8.4 (3.3)
Categorical Data			
Sex, *n* (%) female	115 (69.7)	97 (77.6)	18 (45)
TBW %	53.2 (6.5)	52.3 (4.9)	53.9 (7.5)
ICW %	51.1 (6.1)	50.0 (6.6)	54.5 (2.4)
ECW %	49.0 (6.2)	50.1 (5.6)	45.5 (2.4)
Fracture		125 (75.8)	
Neck of femur	-	50 (40.0)	
Other lower limb	-	40 (32.0)	
Upper limb	-	24 (19.2)	
Pelvis	-	7 (5.6)	
Vertebrae	-	2 (1.6)	
Upper and lower limb	-	2 (1.6)	
Stroke			40 (24.2)
Lacunar	-	-	17 (42.5)
PACS	-	-	5 (12.5)
POCS	-	-	12 (30)
TACS	-	-	6 (15)

Continuous data presented as mean (SD) with the exception of the BUN/Cr ratio presented as median (IQR); categorical data presented as *n* (%). Lacunar, lacunar infarct; TBW, total body water; ICW, intracellular water; ECW, extracellular water; PACS, partial anterior circulation infarct; POCS, posterior circulation infarct; TACS, total anterior circulation infarct.

**Table 2 healthcare-09-00154-t002:** Participant Outcomes.

	All	Fracture	Stroke
Number	165	125	40
Length of Stay	6 (6)	7 (5)	3 (2)
Discharge Destination *n* (%)			
Home	67 (40.6)	38 (30.4)	29 (72.5)
Rehab	63 (38.2)	56 (44.8)	7 (17.5)
Transfer	26 (15.8)	24 (19.2)	2 (5.0)
Died	4 (2.4)	2 (1.6)	2 (5.0)
Missing data *	5 (3.0)	5 (4.0)	0 (0) 0.0

Continuous data presented as median (IQR). Categorical data presented as *n* (%). Good discharge outcome represented by persons discharged home. * Missing data were assumed to be missing at random.

**Table 3 healthcare-09-00154-t003:** Multivariable regression analyses showing relationship between measurements of hydration and outcomes of prolonged hospitalisation and poor discharge destination.

Outcome	BUN/Cr > 15 ^a^		ICW/ECW < 25th Percentile ^b^	
	OR (95% CI)	*p*	OR (95% CI)	*p*
Prolonged hospitalisation ^a^	1.66 (0.60–4.62)	0.33	1.99 (0.90–4.43)	0.90
Unfavourable discharge destination ^b^	3.16 (0.83–12.02)	0.90	4.25 (1.59–11.34)	0.004

^a^ >10 days, BUN/Cr ratio ≤15 as reference; ^b^ rehabilitation, another ward, or death, ICW/ECW ratio >25th percentile as reference.

## Data Availability

The data presented in this study are available on request from the corresponding author.

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
