# Peer review of "Bioelectrical Impedance Versus Biochemical Analysis of Hydration Status: Predictive Value for Prolonged Hospitalisation and Poor Discharge Destination for Older Patients"

_healthcare, 2021, doi:10.3390/healthcare9020154_

Round 1
Reviewer 1 Report
I commend the authors for implementing changes suggested by the reviewers. The manuscript is stronger as a result and reads well.
All of my comments have been addressed. Aside from some minor confusion that still exists with the numbers (line 24, Table1, Table 2), I have no further comments to add.
- Line 24 -- If you present it to be "mean (SD)" I would expect to see "76.7(9.2)". If you want to list it as "76.7+/-9.7", you should lead into it as "mean+/-SD"
- Table 1 -- Having age as "mean(SD)" but then sex below it as "n(%)" is confusing. Yes, you clarify in the caption, but I suggest considering a way to present the values with greater distinction
- Table 2 -- I understand that you are trying to show n-size for discharge destination, but it appears "Discharge Desitnatin (N) %" only to have the subsequent percentages be in parentheses. Minor thing, but a little tweak will help to present the data much cleaner
Reviewer 2 Report
I think the manuscript is clearer, thanks to the authors for the changes. I think this is a novel approach to this topic and will provide the basis for future research.
Reviewer 3 Report
The authors improved the paper following my comments.
The question remains as to whether the use of BIA can actually be an advantage in assessing dehydration in the clinical setting of the hospitalised patient.
As the authors rightly point out in the paper, BUN and creatinine values are routinely available for every patient, as of course is the objective examination. The authors should clarify whether the cost for a hospital to purchase the BIA instrument and to train operators is worth compared to the advantages this instrument offers over the traditional methods mentioned above.
Author Response
Please see the attachment.

This manuscript is a resubmission of an earlier submission. The following is a list of the peer review reports and author responses from that submission.
Round 1
Reviewer 1 Report
In this manuscript, the authors attempted to show that BIA is a useful tool in determining hydration status as it relates to the odds of poor hospital outcomes. The used BUN:Cr is a comparative measure, but had stronger results with BIA. Overall, the paper is well-written, and including BIA, which has been used as a measure in a number of clinical populations, in a new population has merit.
Line 24 -- You say "(mean age (SD)" but then follow up with "76.7+/-9.2". Edit for consistency.
Line 79 -- Why were two different BIA devices used? I understand the testing was done in separate locations, but varying equipment/manufacturers leads one to wonder how consistent the devices are with one another. Has separate analyses been performed to show the two BIA devices are consistent with one another?
Line 111-114 -- You have "mean (SD)", "median (IQR)" and "n(%)" in the table. I suggest altering the appearance of the data as "value x (value y)" shown throughout would lead one to believe the presentation of data is the same, when it is not. Also, if median/IQR is presented for BUN, I assume BUN:Cr was not normally distributed. Any additional statistical operations in this case? If so, that needs to be stated in Section 2.3.
Line 119 -- Similar comment to Table 1 regarding "value (value)" duplication.
Line 120 -- Table 2 caption says "Categorical data presented as n(%)" however, the table appears to show it as "% (N)". Edit for consistency.
Line 140 -- The manuscript seems to tout BUN:Cr as an objective "gold standard" for clinical measuring dehydration status. However, the statement "appears to be a better prognostic indicator than BUN:Cr". While BIA may have indeed shown better indication of poor outcomes, the statement leads one to think BIA may actually be measuring something other than what BUN:Cr is supposed to measure. Please address.
Line 149 -- "We propose that BIA could be used to guide patient management rather than as a gold standard measurement of dehydration...". Similar to the previous comment, it appears that BIA and BUN may be measuring different constructs. I venture the original hypothesis of this study was for BIA and BUN results to be similar, so the argument could be made for BIA to be a quicker and effective tool. However, if only BIA is showing significant relationship with your outcome data, it is hard to make such a conclusion. Please address.
Line 160 -- The use of "out" in this sentence seems incorrect.
Line 173 -- "...at risk of adverse outcomes..." is extended hospital stay not an adverse outcome? If it is, then the tool was not strong in that regard.
Reviewer 2 Report
Page 1, Line 26 - must define BIA - I found the definition later in the body of the manuscript but it would need to be defined in the abstract as well, as this often stands alone.
Page 1, Line 33-34 - is dehydration the commonest fluid disorder in general or just among older patients? is dehydration associated with increased morbidity and mortality in older patients compared to younger patients with dehydration or compared to older patients without dehydration?
Page 1-2, Line 41-42 - thirst is listed as a sign, but it is really a symptom reported by the the patient - consider rephrasing to "signs and symptoms"; orthostatic drop in blood pressure is listed as a clinical sign, and then grouped as something subjective in the next line - blood pressure changes are measurable and not subjective - consider rephrasing to something like "clinical diagnosis may be subjective" since some clinical signs are not subjective.
Page 2, Line 48 - "in health" is very vague, consider rephrasing to something like "in healthy patients" or "in patients with normal hydration"
Page 2, Line 59 - clarify what you mean by "discharge to an unfavourable destination" - does this mean something like a rehab facility versus home? I see it finally defined on Line 89, but it needs to be explained in general terms sooner.
Page 2, Line 66 - "The present study therefore included 165 participants (mean age (SD) = 76.7±9.2 years)." - this should be in the results section, not the methods
Page 2, Line 67-72 - you state in the abstract "a cohort of prospectively
18 identified elderly (>65 years) patients" and throughout the introduction you reference this age group (>65 years), yet when we come to the methods, stroke patients were included who were >50 years - you need to be clear in the abstract and throughout that your study group was NOT just "older patients (>65 years)" and WHY.
Page 2, Line 71 - need a reference for NIHSS
Page 2, Line 78-80 - can you explain why you used difference equipment on the different patient populations? can you describe the equipment in similar terms (for example, the frequency)?how might this have affected the results? if you combine this with the difference age groups in each patient population, how can you consider the results collectively?
Page 3, Line 92-102 - did you calculate the sample size that would be needed? you should include the basics details about the statistics (not just the software used, but the actual numbers), so the reader can confirm whether the findings were significant
Page 4, Table 2 - "missing" suggests the patient is missing, but I think you mean the patient was lost to follow up or the outcome was unknown? this should be clarified. It's also confusing that you present the specific destinations of where patients were discharged to and then you present them collectively as good versus poor outcome, when you already defined that poor outcome collectively included anyone not discharged to home. This second part of the table is redundant and I suggest removing.
Page 1, Line 59 - you state that the cohort is prospectively identified, but on Page 4, Line 136 - you state that this analysis is retrospective. If you prospectively identified and followed these patients, then this is a prospective analysis.
Last line of abstract: "BIAr could be used utilised as a fast, reliable and non-invasive method for detecting dehydration at the bedside." - sounds like you are suggesting using this tool alone, whereas in the conclusion, you are suggesting using this tool in combination with other tests and methods of clinical assessment - need to revise the abstract to make it in sync with the conclusion
Last paragraph of discussion, you discuss limitations, but no where do you mention using different equipment, and if this is not a limitation, you need to explain why.
In the conclusion, you again return to "older patients". You need to clearly state throughout the paper that your study group included patients >50 years and therefore you cannot just apply the results to patients >65 years.
Punctuation was confusing throughout:
- Use of parentheses within parentheses
- examples on Page1, Line 20-21, 24, 25
- Absence of commas and/or use of "and" to join more than two subjects or objects
- example on Page 1, Line 19-22
- example on Page 5, Line 153-154
- Run-on sentences
- example on Page 2, Line 51-53, 53-56
- Variation in how parentheses are used
- Page 2, Line 43: "BUN to Cr (BUN:Cr) ratio"
- Page 2, Line 47-48: "intracellular water (ICW) to extracellular water (ECW) ratio (ICW:ECW)"
Reviewer 3 Report
Aim of this study was to compare the performance of ICW:ECW ratio calculated through BIA to BUN:Cr ratio in predicting poor outcomes in a cohort of elderly patients.
The study offers interesting perspectives for possible clinical implications and for the development of new studies on the subject.
However, some changes are necessary before publication.
- Data were drawn from two patient cohorts with different clinical conditions. Two different BIAs were used. Could this affect the result? Wasn't it preferable that the patients were just one group?
-
Results from multivariable logistic regression analyses examining the relationship between measures of dehydration and study outcomes showed some correlations. Confounding factors such as age have not been considered. Is it possible that the model does not offer exact results?
- The discussion of the work is rather concise. Comparisons should be added to the reader's reading on the subject and the possible uses of BIA in this type of patient.